# Quantifying the Displacement of Data Matrix Code Modules: A Comparative Study of Different Approximation Approaches for Predictive Maintenance of Drop-on-Demand Printing Systems

**DOI:** 10.3390/jimaging9070125

**Published:** 2023-06-21

**Authors:** Peter Bischoff, André V. Carreiro, Christiane Schuster, Thomas Härtling

**Affiliations:** 1Fraunhofer Institute for Ceramic Technologies and Systems IKTS, Maria-Reiche-Str. 2, 01109 Dresden, Germany; christiane.schuster@ikts.fraunhofer.de (C.S.); thomas.haertling@ikts.fraunhofer.de (T.H.); 2Institute of Solid State Electronics, Technische Universität Dresden, 01069 Dresden, Germany; 3Senodis Technologies GmbH, Manfred-Von-Ardenne-Ring 20 D, 01099 Dresden, Germany; 4Fraunhofer Portugal Center for Assistive Information and Communication Solutions—AICOS, 1649-003 Lisbon, Portugal; andre.carreiro@aicos.fraunhofer.pt; 5Fraunhofer Portugal Center for Smart Agriculture and Water Management—AWAM, Rua Alfredo Allen 455/461, 4200-135 Porto, Portugal

**Keywords:** data matrix, pattern recognition, drop deviation, code recognition, quality assessment, predictive maintenance

## Abstract

Drop-on-demand printing using colloidal or pigmented inks is prone to the clogging of printing nozzles, which can lead to positional deviations and inconsistently printed patterns (e.g., data matrix codes, DMCs). However, if such deviations are detected early, they can be useful for determining the state of the print head and planning maintenance operations prior to reaching a printing state where the printed DMCs are unreadable. To realize this predictive maintenance approach, it is necessary to accurately quantify the positional deviation of individually printed dots from the actual target position. Here, we present a comparison of different methods based on affinity transformations and clustering algorithms for calculating the target position from the printed positions and, subsequently, the deviation of both for complete DMCs. Hence, our method focuses on the evaluation of the print quality, not on the decoding of DMCs. We compare our results to a state-of-the-art decoding algorithm, adopted to return the target grid positions, and find that we can determine the occurring deviations with significantly higher accuracy, especially when the printed DMCs are of low quality. The results enable the development of decision systems for predictive maintenance and subsequently the optimization of printing systems.

## 1. Introduction

### 1.1. Print Quality Assessment in the Context of Predictive Maintenance

Individual part marking is a technology that enables the digitization of production processes. It allows the identification of single components, e.g., via unique identifiers (UID), and, hence, facilitates assigning production parameters to these individual parts. Based on these part-specific data, several measures for process optimization can be derived. One option to carry out part marking is drop-on-demand printing (DoD, e.g., for barcodes or data matrix codes (DMC)). If special colloidal inks containing glass and ceramic particles are utilized, multiple unpublished experiments and use in real-world industrial applications have shown that the printed codes can withstand the high temperatures used in hot forming processes in the metal industry.

An important aspect in metal processing is that marking a part means adding a quality parameter to the component. If the print is not readable, the part does not fulfill its quality specifications and, hence, must be considered scrap. This means that the printing process requires constantly high quality and the printer must induce a minimum downtime. These requirements can be met through the implementation of continuous monitoring and predictive maintenance strategies for the printer.

In [1], we presented a new way to assess the printing system’s state for maintenance planning, solely through the analysis of the printed pattern instead of hardware monitoring. Our method is based on drawing conclusions concerning the remaining useful lifetime (RUL) of the printing system from the temporal evolution of printing errors. This primarily includes the deviation of individual printed dots from their target position.

The target positions (*T*) describe the center coordinates of the binary data cells (the so-called modules) of an ideal DMC, which is perfectly aligned in a quadratic grid with all rows and columns evenly distanced. The effect of print quality degradation goes back to, e.g., partial nozzle clogging in the print head, which can induce different characteristic deviations from the ideal quadratic printing result. In this work, we consider the following two positional errors:(1)The constant deviation of all modules in a column in the same direction from the target position. This represents a misalignment of all ejected ink dots from one nozzle of the print head.(2)The so-called fading shift of the first few modules in a column. In this case, the deviation from the target position is at its maximum for the first printed dot and gradually decreases over the next dots in the same column. This behavior represents a self-cleaning effect of the nozzle during printing. These errors are displayed in Figure 1 below.

In order to evaluate the temporal evolution of the printing quality and, hence, the status of the printer, the deviation between the dots actually printed at positions *P* and the target positions *T* needs to be calculated, while the set of positions *P* can directly be retrieved from the recorded image of the DMC. The challenge of this approach lies in the fact that the target positions *T* are a priori unknown. They need to be approximated from the printing pattern itself. In this paper, we report our approach to solving this problem computationally, which finally leads to an early quantification of printing errors not easily noticed with the naked eye.

### 1.2. Literature Review

To our knowledge, there is no previous work on analyzing drop-on-demand printing in the described way and therefore also no previous work on how to precisely approximate the target positions. In general, little work has been carried out on detecting and analyzing DMCs. Few authors have contributed works towards recognizing and localizing codes in images. Huang et al. presented an approach based on line segment detection and border fitting by minimizing the distance of border points to the detected lines [2]. Other authors utilized Hough transforms or the more specialized Radon transform to detect lines and recognize the respective code symbols [3,4]. An approach that uses machine learning (support vector machines) was published by Cho et al. [5]. Karrach and Pivarčiová published an extensive comparison of different methods to localize data matrix codes and compared their performances to their own method [6].

Furthermore, a number of standards exist that describe methods of detecting and decoding DMCs and two-dimensional codes, respectively. ISO 15415 describes a standard methodology for grading and evaluating measurements of specific attributes of two-dimensional symbols [7]. This standard also describes the five main tasks of decode algorithms: localization of the symbol, determination of reference points, definition of the nominal grid, error correction, and decoding the symbol. In our context, the third task is of especially high interest. The decoding algorithms have in common that they define the center positions for the set of modules and evaluate the contrast of the area around this center (the aperture) in comparison to the entire symbol area to evaluate the binary value of each module. One reference decode algorithm for DMCs is described in ISO 16022 [8]. It fulfills the tasks defined in [7]. The main steps of this algorithm are to detect the edges of the DMC symbol and to follow the edges that cross a horizontal or a vertical scan line to filter contours meeting certain criteria to identify the L-shaped finder pattern. Through this pattern the orientation of the DMC is defined by setting all modules (compare Figure 2: left column and top row in both DMCs). Given the outer edges of the L-shaped finder pattern, two more scan lines, parallel to the sides of L-shape, are defined and shifted until the transitions along the scan lines on both sides match to identify the timing pattern, where only every second module is set.

The reference decode algorithm from [8], like the other algorithms, is designed to work with DMCs printed in such a way that neighboring modules are connected to each other (Figure 2a). ISO 29158 [9] includes modifications to the definitions made in [7] and also includes an algorithm to connect dots that are separated by a distance of less than one module width, while not connecting areas that are separated by more than the module width.

The accuracy of the method used to approximate the target positions directly impacts the maintenance system of the printer. Therefore, we aim to introduce different algorithms, compare their accuracy, and investigate their limits. We compare our algorithms to an adaption of the code recognition method from Karrach and Pivarčiová [6], which we adapt to codes printed with individual dots (Figure 2b) by first connecting the dots with the methods defined by the International Standardization Organization in [9]. Finally, we select one of the algorithms based on their performance but also considering the computational efficiency and simplicity of implementation and suitability for a broad variety of use cases.

Our work contributes to building a predictive maintenance system, which is based purely on the printed image and does not require additional sensors and monitoring of the actual print head. Therefore, both the necessity and the effect of maintenance actions can be monitored with higher relevance due to the close relation to decodability.

## 2. Materials and Methods

### 2.1. Algorithms

We compare five different algorithms to approximate the target positions of dot-printed DMCs. The first algorithm, *Adapted Connected Data Matrix Code Recognition (CDCR)*, is based on the best method published in [6]. The remaining four algorithms are categorized according to both whether or not the algorithm requires any previous knowledge about the printed symbol (e.g., the number of rows and columns), and the underlying mathematical method, either based on clustering algorithms or on affine transformations.

To equalize the conditions for all algorithms and achieve comparable results for the experiments with real images, we preprocess the raw images of the DMCs. The first preprocessing step is to segment the ink dots from the background. Given the center coordinates of each ink region from the segmentation, we perform a rotational adjustment by rotating all coordinates stepwise by 0.5 degrees until the bounding box that includes all coordinate points is minimized.

#### 2.1.1. Adapted CDCR

The algorithm described in [6] works on standard DMCs printed with connected modules, as shown in Figure 2a. To apply this algorithm to our problem given the center coordinates of each segmented ink dot without making significant changes to the algorithm, we first have to create a suitable input image. Therefore, we first create circular areas with a predefined radius on a black background and then apply the dot-connecting algorithm from ISO 29158 [9]. We point the reader towards the original publication of this algorithm to understand more details and will only explain a few adaptations we made.

Karrach and Pivarčiová filtered the connected regions according to their size, their ratio of width and height, and their extent. To find the L-shaped finder pattern from the remaining regions, the authors searched for a region where at least three points formed the vertices of an isosceles right-angled triangle. We replace this step by searching for the region with the longest distance between any two points that are part of the respective region. Due to the high resolution and the characteristics of our input images, partly caused by artifacts from the dot-connecting algorithm, the original filtering was inconclusive.

Since we have extensive control of the imaging process and we do not simulate any perspective distortion, we refrain from implementing the perspective distortion identification step and continue with identifying the timing pattern. We identify the timing pattern by copying the line segments defining the outer edges of the finder pattern and shifting the copied segments towards and over the image center until they do not cross any foreground pixels. We then shift the line segments back pixel-wise until we find the largest overlap with the foreground, which gives us roughly the center of the timing pattern lines. The number of transitions on the timing pattern determines the number of rows and columns. In the final step, the grid is defined as a regular coordinate matrix with equal row and column distances.

#### 2.1.2. Determining the Number of Rows and Columns in an Unknown DMC

Since our algorithms do not rely on finding the timing pattern to find the dimension of the DMC, we find the possible dimensions by counting the number of positive modules in the DMC. To find the limits to determine the dimensions, we created 100 random strings, including uppercase, lowercase, digits and punctuation, of up to 40 characters, encoded using the DMC standard, and evaluated the number of positive modules in each DMC. This resulted in the limits shown in Table 1. If a DMC should fall into two possible dimensions, we analyze it twice and use the result with lower deviation values.

#### 2.1.3. Clustering Algorithms

Approximating the target positions *T* using clustering approaches is based on the idea that the deviation of the individual dots has a centered distribution. We create clusters of points by projecting the entire set of center coordinates of the printed points *P* to either the lowest column coordinate (to identify the row positions) or the lowest row value (to identify the column positions). Different established clustering algorithms are tested to identify the centers of the thereby formed clusters. We experiment using the mean-shift algorithm [10,11] and the K-means algorithm [11,12,13]. The mean-shift algorithm’s independence from a priori knowledge of the number of clusters and its performance working with uneven cluster sizes exhibits a possible advantage when applied to DMCs with large deviations. This would be attributed to the fact that some center coordinates might actually be closer to the next row or column than to their respective target row or column.

Following the identification of cluster centers, the target coordinates are defined as the combination of row cluster centers and all column cluster centers. This creates a grid of regularly distributed coordinates, which we refer to as the set of approximated target positions *T*.

#### 2.1.4. Affine Transformations

A different approach to approximate the target positions *T* is described in the following section. This approach also starts with the given coordinates of the printed modules *P*. Given that an ideal data matrix code is square with the same constant distance between each row and column, we can create a preliminary set *T* as a regular distributed grid of coordinates with the required number of rows and columns. To find an appropriate seed value for module size (equal to the distance between two consecutive rows or columns, respectively), we use the maximum distance of two points in one dimension divided by the approximated (see Section 2.1.2) or given dimension.

We then optimize *T* by minimizing the distance of each point in *P* to the closest point in *T* over a set of affine transformation operations, viz. translation, rotation, and scaling operations. We have shown in [1] that there are certain types of errors to be expected when printing colloidal inks with the DoD technology. One of these errors is characterized by a fading deviation of the first few dots from each nozzle (typically three to four dots). Given this knowledge, we also compare the optimization results considering this error by applying a lower weighting to dots with a higher probability of being affected by the fading deviation in the objective function. To determine the affected dots correctly, we recreate the DMC matrix by either decoding the string from *P* and re-encoding it instead of creating the full grid or by reading the string from our database. The latter would of course only be an option in a production environment if the DMCs are analyzed by the same party that is responsible for printing.

### 2.2. Experiments

The following section describes our experimental procedure to compare the different algorithms in order to find the best match between the approximated target positions $T_{approx}$ of each dot (used to compute the deviation of each printed dot) and the real target positions *T*, which are only known in printing simulations, and to evaluate the results on real data from long-term printing tests. We start the comparison by performing tests to determine a comparable runtime and find performance differences in the different methods of approximating the target positions. This is performed on a workstation equipped with a 4.5 GHz CPU (Intel Core i9-10900X) by running each algorithm a hundred times and comparing the mean computation time as well as the standard deviation. We are only comparing the differences, not the absolute results, since the implementations are not focused on performance at this point.

#### 2.2.1. Simulation of Printing Errors

Quantifying the results of the individual algorithmic approximations of the target positions requires knowledge about the real target positions, since it is insufficient to compare approximated targets (*T*) to the printed positions (*P*). Therefore, we simulate printed DMCs with the known failure modes from [1], which are constant column shifts (a constant shift of all dots produced by one nozzle of the print head in the same direction), fading column shifts (the directional shift of dots fades with the printing direction, typically the first three or four dots), and the general positional uncertainty of all dots.

Using the simulated printing, we determine the impact of these three errors and ensure that the results are independent of the encoded string and therefore the exact positions in the DMC grid, which are printed or not. For the simulation we create square DMCs with 16 rows and columns. We encode seven different strings and of each DMC up to seven columns (out of the 16 columns) are manipulated to contain either both or one of the shifting error modes. Additionally, one of seven different mean values is used to generate the general positional uncertainty distribution, which is added to the positions *P* of each simulated DMC and superposed with the shifting error modes. Each combination is created ten times, resulting in 31,360 simulated DMC position arrays, which are used to compare the algorithms. The quality of the approximation is quantified by the relative number of positions in the DMC that is approximated correctly within a certain threshold (e.g., within a five-pixel threshold away from the center of the module with a module size of 30 pixels, which we named $t_{1/6}$). Figure 3 displays how a single string is encoded as a DMC and how the algorithms we describe are applied to evaluate their performance in matching the approximated target positions $T_{approx}$ to the real target positions *T*.

#### 2.2.2. Time Series

In addition to comparing the individual algorithms quantitatively, we apply the algorithms to the printed positions segmented from real DMC prints by segmenting the dots from the background and applying connected component labeling (compare [1,14]). The DMCs are printed in experiments on our test bench. We produce two time series of DMCs with 92 and 63 prints, respectively. All prints from both time series are analyzed individually before the mean deviation values computed through the individual algorithms are plotted for comparison. The time series are compared qualitatively to find any irregularities that either do not match the visual impression of the printed DMCs or are caused by the wrong behavior of the algorithms towards segmentation failures. The flowchart in Figure 4 shows how the images of the printed DMCs are preprocessed to find the coordinates of the printed positions *P* before approximating the target positions *T*.

## 3. Results and Discussion

### 3.1. Runtime Performance

Even though runtime is not the primary decision factor used to determine which algorithm is used for real-time maintenance information, it still gives an important insight into the suitability of the algorithms. Both the clustering approach (C) and the affine transformation approach (AT) can be combined with additional information if available to guide the algorithm towards finding the correct target positions. We named the algorithms accordingly as unguided and guided affine transformation (UGAT and GAT) and unguided and guided clustering (UGC and GC). Figure 5 shows clear differences in how quickly the target positions can be approximated by the algorithms based on affine transformations and clustering, respectively, as well as a comparison to the adapted CDCR. It is important to note that the implementations in this work are not optimized for performance. The comparison only serves to show relative differences. It clearly indicates that the algorithms based on affine transformations have a better performance than the algorithms based on clustering and that both methods outperform the adapted version of the algorithm by Karrach and Pivarčiová.

### 3.2. Theoretical Performance Evaluated through Simulated Prints

Before comparing the algorithms on drop-on-demand-printed DMCs, we compare them theoretically using simulated prints with known position grids and applying the error modes previously described at different strengths. This allows us to compare the approximated targets to the position grid, which represents the actual target positions, and therefore to quantify the results accurately. Figure 6 displays how each algorithm is capable of approximating the correct target positions when faced with either a significant amount of fading shift errors (a) or constant shift errors (b). For this visualization, only simulations with seven columns impacted by one or the other error and a general positional uncertainty of ten percent of the module size were chosen. The results clearly indicate that the algorithms have different capabilities when it comes to dealing with these errors. The affine transformation algorithms are less affected by both the fading shift and the constant shift error when compared to the clustering-based methods. The unguided affine transformation clearly shows a greater number of matches in the case of the constant shift error. Even though the results are not as clear for the fading shift error, due to the simple implementation, which does not require any additional knowledge, the unguided affine transformation (UGAT) is preferred over the guided affine transformation algorithm.

If all simulations described above are taken into account, i.e., including combinations of both shifting error modes, this tendency continues (Figure 7): in comparison to the adapted CDCR algorithm, the UGAT algorithm shows a much better performance. This includes simulated DMCs where both the fading and the constant shift error are apparent and combined with larger general positional uncertainty than those in Figure 6. This explains why the difference in performance is even more clear than in the previous comparison.

It is easily noticed that even though cases exist where the CDCR algorithm matches all target positions, there are many cases where only an insufficient amount of positions is matched at this threshold. This shows that even though the algorithm is highly capable of decoding even low-quality DMCs, the requirements for decoding differ from what we require to assess the quality of the prints and state of maintenance of the print head, respectively.

### 3.3. Comparison of Time Series of DoD-Printed DMCs

Last but not least, the algorithms are compared on DMCs printed using the drop-on-demand technology. This is important as the preprocessing steps (localization and segmentation) can introduce additional errors and uncertainties, which have to be compensated for by the algorithms. It is furthermore important to note that we do not know the actual target positions in this experiment and can therefore only compare the results qualitatively by observing the time series.

The time series printed in this case does not show any time-dependent behavior according to the expert evaluation and our previous method of analysis. As a consequence, it is easier to compare the time series and find differences as we do not have to compare any rising trends, possibly measured with different increases. As seen in Figure 8, the time series differ significantly. We do not consider the absolute values of the individual prints but rather only the characteristics of the series and especially the variation over time.

It is clear to observe that the clustering approaches produce a signal with greater variance and noise. The affine transformation approaches are similar both in absolute values and in their evolution. Having a smoother signal with less noise enables more confident conclusions to be made about the maintenance state of the system.

## 4. Conclusions

In this paper, we present an in-depth analysis on how to accurately quantify the printing quality of DMCs produced with a DoD printing system. The quality of the evaluation depends on the quality of the underlying data. Therefore, our results are an important step towards the practical implementation of a predictive maintenance system for DoD printers. We investigate the performance of a number of different algorithms to approximate the dot target positions. The results were compared quantitatively for simulated data matrix code positions and qualitatively for real time series of printed data matrix codes to validate the theoretical results and evaluate the consistency with real-world data.

We show that the choice of algorithm used to measure the positional deviation of the individual dots from the DoD printing technology has a large impact on the measured printing quality. Even though detecting the rows and columns by clustering algorithms works well for DMCs with little to no deviation, the quality of the approximation decreases when there are more errors present. Overall, using affinity transformations results in more accurate approximations of the target positions with, in most cases, almost 100 percent of positions matched within the applied threshold, whereas the adapted decoding algorithm in many cases only matches a small number of positions with the same threshold. The affinity transformations additionally result in more consistent evaluations of the printing quality when applied to real-world data.

We conclude that even though the results of the affinity transformation can in certain cases be increased by using additional knowledge, e.g., the encoded string, we generally achieve sufficient results without additional knowledge. This makes our approach suitable for a wide range of applications, including those where no additional information is available. We are currently working on implementing these results in an extensive software environment for continuous system monitoring and the collection of data to build a production-integrated decision system.

## Figures and Tables

**Figure 1 jimaging-09-00125-f001:**
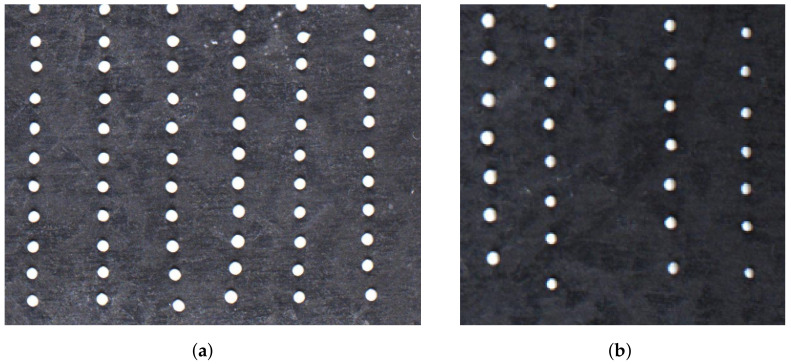
Visualization of the different positional error modes on a specially printed test pattern of 16 columns and 20 rows (partially displayed): a fading shift of the first few dots (e.g., third and fourth column from left) (**a**) and a constant shift of one or more columns (**b**).

**Figure 2 jimaging-09-00125-f002:**
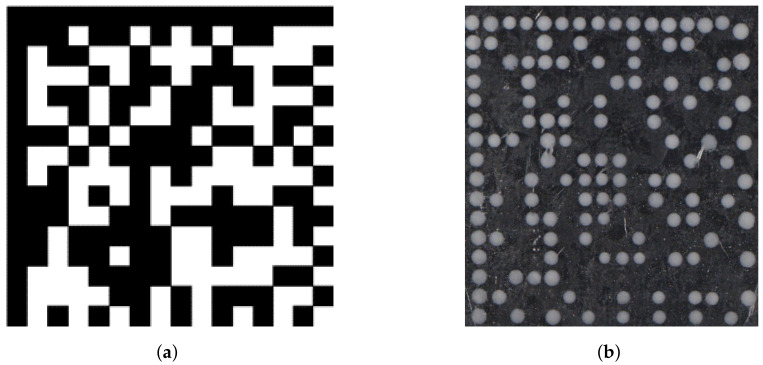
(**a**) A standard data matrix code with connected modules. (**b**) A dot-printed data matrix code printed with the DoD process analyzed here.

**Figure 3 jimaging-09-00125-f003:**
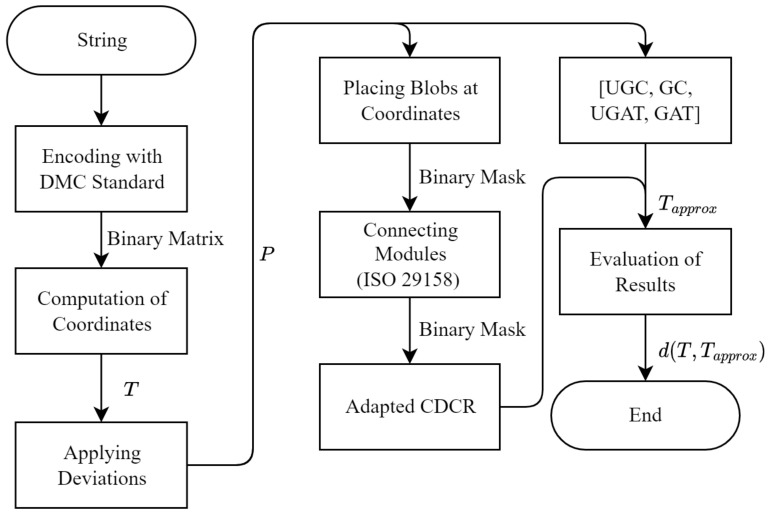
Flowchart of the methodology for the performance evaluation of our algorithms and comparison against the Adapted CDCR algorithm. After creating the synthetic DMCs with the target positions *T*, deviations are applied for the printed positions *P*. The algorithms are evaluated by the distance between the approximated target positions $T_{approx}$ and the actual target positions *T*.

**Figure 4 jimaging-09-00125-f004:**
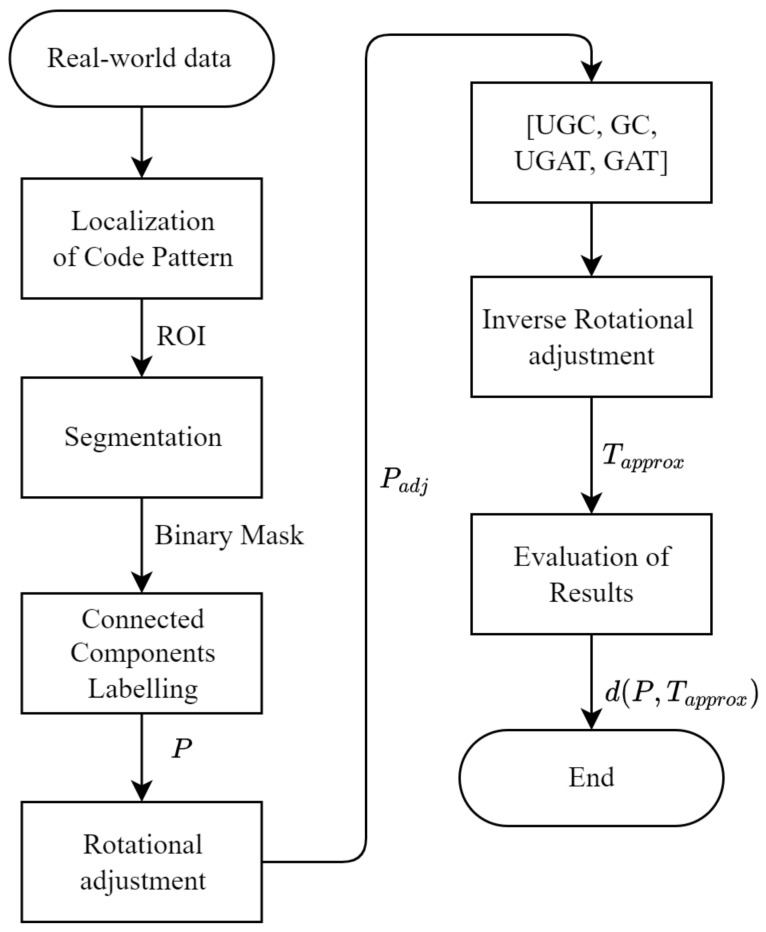
Applying the algorithms to actually printed DMCs after preprocessing by localizing the region of interest (ROI) and segmenting the ROI to find the center positions of each blob. The printed positions *P* are rotated until the smallest bounding box is found to ensure that the DMCs finder pattern is parallel to the image axes. After applying the algorithms, the found positions are rotated by the inverse rotation matrix from the previous step to match *P*.

**Figure 5 jimaging-09-00125-f005:**
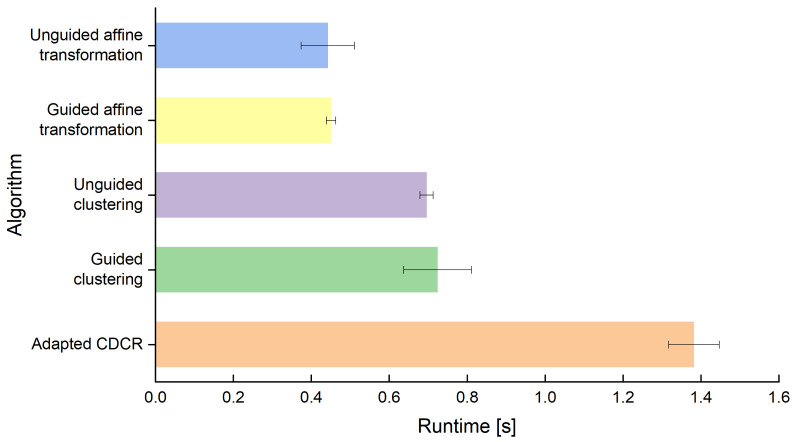
Runtime of the compared algorithms for a single data matrix code averaged over 100 runs, including the standard deviation of the runtime.

**Figure 6 jimaging-09-00125-f006:**
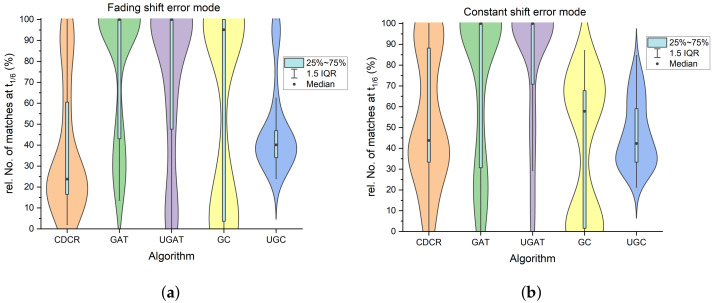
Distribution of the number of matches of real target positions and approximated target positions *T* after applying a threshold of 1/6 of the module size for each of the tested algorithms. (**a**) shows the distributions considering only the fading shift error while (**b**) shows the distributions of matches when analyzing only the constant shift error. The blue bar displays the range between the first and the third quartile. The whiskers span over 1.5 times the interquartile range (IQR) (Q3−Q1) and the outer contour is approximated through kernel density estimation (KDE).

**Figure 7 jimaging-09-00125-f007:**
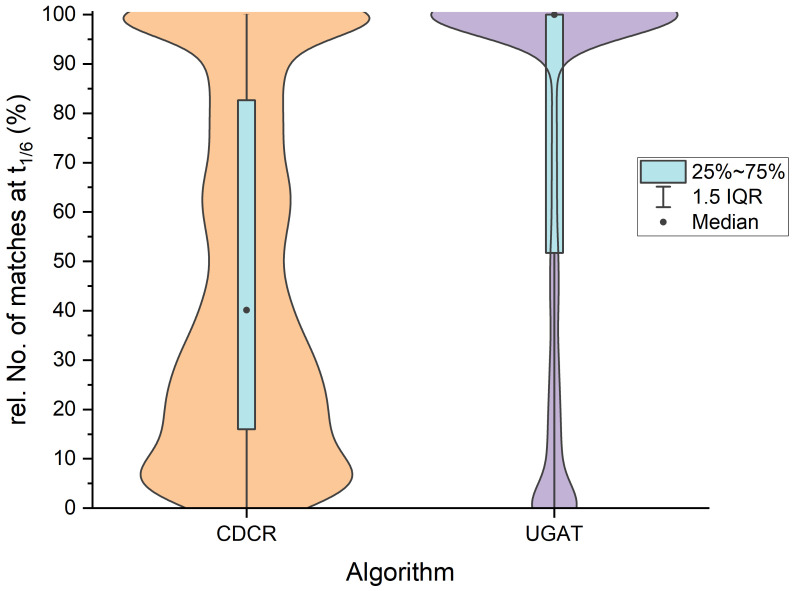
Comparison of the best-performing algorithm from our work to an adapted version of the best-performing algorithm with respect to decoding DMCs from [6], over all simulations described in Section 2.2.1.

**Figure 8 jimaging-09-00125-f008:**
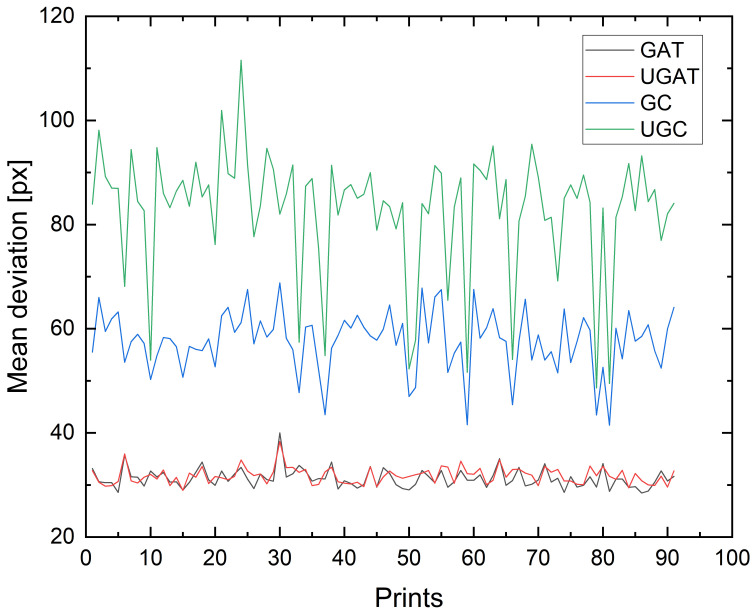
Comparison of the time series analysis by both versions of both the clustering- and the affine-transformation-based methods. The differences in absolute values are not of as much interest as the variance along the time axis.

**Table 1 jimaging-09-00125-t001:** Stochastically determined limits of the number of positive modules in a data matrix code to find the dimensions of the data matrix code from given dot coordinates.

Dimension	Lower Limit	Upper Limit
8	20	41
10	32	58
12	53	84
14	79	112
16	103	146
18	128	180
20	165	227

## Data Availability

The source code generated within in this study is openly available under the MIT license at https://gitlab.cc-asp.fraunhofer.de/ikts-oss/gridfinder. The data presented in this study are available from the author upon request.

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
