# Peer review of "Quantifying the Displacement of Data Matrix Code Modules: A Comparative Study of Different Approximation Approaches for Predictive Maintenance of Drop-on-Demand Printing Systems"

_2313-433X, 2023, doi:10.3390/jimaging9070125_

Round 1

Reviewer 1 Report

Abstract-the originality and novelty of the research activity are not well exposed and highlighted.

Page 2-3, The Literature review has to be improved. The review is poorly written and missed some important milestones in the presented research.

Page 3, Chapter 2. Materials and Methods, 2.1 Algorithmus- please describe your selected methods. Pro and Contra for methodology.

Page 10, References- add the list of the relevant latest literature

The presented manuscript is clearly written.

Reviewer 2 Report

This paper presents a detailed analysis of how to accurately measure the printing quality of DMCs produced with DOD printing systems. The authors compare the performance of different algorithms to approximate dot target positions and evaluate the consistency of the theoretical results with real-world data. They conclude that using affinity transformations results in more accurate approximations of target positions and consistent evaluations of printing quality. Overall, the paper provides valuable insights into the practical implementation of a predictive maintenance system for DOD printers. However, I have some questions to the authors:

1) A detailed description of the algorithms compared in Figure 2 is missing. Flowcharts or references to papers where these algorithms are described in detail should be included. 

2)The introduction does not describe the problems very clearly, it is necessary to give examples of poor printing and describe the problems that cause defects. At the end of the introduction, add the main contribution of the article.

3) The fourth section should be titled "Conclusion", and this section should include an explanation of the scientific novelty, the main results, and some numerical estimates of the results obtained.

Minor issues:

1) Sentence in line 23-25 need a reference on literature.
2) All abbreviations in Figure 3 must be explained in the text.(GAT, UGAT, GC, UGC)
3) Line 243. "Figure 2 displays..." probable Figure 3?
4) Decide on the abbreviation DoD or DOD.

Reviewer 3 Report

The present work proposes a Comparative Study of Different Approximation Approaches for Predictive Maintenance of Drop-on-Demand Printing Systems. Here are the main comments about the manuscript which need to be considered:

a) The authors should describe, in detail in the materials and methods chapter, the algorithms used, i.e., in terms of mathematical description.

b) It is also worth considering the entire research procedure used by the authors in the form of, for example, a graphic diagram. This will allow for a more transparent presentation of the research.

c) The authors of the article presented a simulation of the presented algorithms. Did the authors undertake the implementation of the algorithm and its verification during the printing

The paper is reasonably properly written (but the article needs a typo-grammatical check).
